# The Relationship Between Thrombophilia and Modifications in First-Trimester Prenatal Screening Markers

**DOI:** 10.3390/medicina61020318

**Published:** 2025-02-12

**Authors:** Viorela Romina Murvai, Casandra-Maria Radu, Radu Galiș, Timea Claudia Ghitea, Anca-Florina Tătaru-Copos, Alexandra-Alina Vesa, Anca Huniadi

**Affiliations:** 1Doctoral School of Biological and Biomedical Sciences, University of Oradea, 1 University Street, 410087 Oradea, Romania; rominna.cuc@gmail.com (V.R.M.); rcasandra1996@gmail.com (C.-M.R.); ancutza_copos@yahoo.com (A.-F.T.-C.); ancahuniadi@gmail.com (A.H.); 2Department of Obstetrics and Gynecology, Emergency County Hospital Bihor, 65 Gheorghe Doja Street, 410169 Oradea, Romania; 3Department of Neonatology, Faculty of Medicine and Pharmacy, University of Oradea, 1 University Street, 410087 Oradea, Romania; raduoradea@yahoo.co.uk; 4Pharmacy Department, Faculty of Medicine and Pharmacy, University of Oradea, 1 University Street, 410087 Oradea, Romania; 5Department of Morphological Sciences, Faculty of Medicine and Pharmacy, University of Oradea, 1 University Street, 410087 Oradea, Romania; 6Department of Surgical Sciences, Obstetrics and Gynecology, Faculty of Medicine and Pharmacy, University of Oradea, 1 University Street, 410087 Oradea, Romania

**Keywords:** thrombophilic mutations, first-trimester screening, parity, PAPP-A, nuchal translucency

## Abstract

*Background and Objective*: Thrombophilia significantly increases the risk of complications like recurrent pregnancy loss, preeclampsia, IUGR, and stillbirth. Objective: This study aimed to evaluate the impact of inherited thrombophilic mutations on first-trimester screening outcomes, focusing on their relationship with maternal biomarkers and ultrasonographic parameters. *Materials and Methods*: A prospective observational study was conducted on 105 pregnant women during the first trimester (10–13 weeks of gestation). Genetic testing identified common thrombophilic mutations, including factor V Leiden, prothrombin G20210A, and MTHFR polymorphisms. First-trimester screening parameters, including PAPP-A, free β-hCG, and nuchal translucency (NT), were assessed. Maternal demographic and clinical characteristics, such as parity and smoking status, were recorded. Pearson correlation and risk estimates were calculated to explore associations between thrombophilic mutations, maternal factors, and screening results. *Results*: Lower parity (≤2) was significantly associated with a reduced risk of low PAPP-A levels (<1.0 MoM) (OR = 0.173; 95% CI: 0.044–0.676). Non-smokers showed a trend toward lower risk of low PAPP-A, although the association was not statistically significant. NT measurements <2.5 mm were consistent with normal fetal development, while maternal factors such as chronic hypertension and a history of small-for-gestational-age infants showed no significant correlations with screening markers. No significant association was observed between thrombophilic mutations and biomarker levels. *Conclusions*: Parity emerges as a significant factor influencing first-trimester screening outcomes, particularly PAPP-A levels, underscoring the need for tailored risk assessments in multiparous women. While smoking and thrombophilic mutations showed no definitive impact, their potential role in placental dysfunction warrants further investigation. These findings emphasize the importance of integrating maternal characteristics into screening protocols to enhance predictive accuracy and maternal–fetal outcomes.

## 1. Introduction

The first trimester of pregnancy represents a critical stage marked by rapid embryonic development and the establishment of placental architecture. During this period, first-trimester screening has become a cornerstone of prenatal care, providing an early evaluation of the risk for chromosomal abnormalities and adverse pregnancy outcomes. This screening process integrates biochemical markers, such as free beta-human chorionic gonadotropin (β-hCG) and pregnancy-associated plasma protein-A (PAPP-A), with ultrasonographic measurements, most notably nuchal translucency (NT). These markers reflect the health of both the fetus and the placenta, which are essential for sustaining a healthy pregnancy [1,2,3,4,5].

Despite the clinical utility of first-trimester screening, maternal factors, including genetic predispositions, can influence the results and complicate their interpretation. Among these factors, inherited thrombophilias are of particular concern. Thrombophilias are genetic or acquired conditions that predispose individuals to thromboembolic events. The most common inherited thrombophilias include the factor V Leiden mutation, prothrombin G20210A mutation, and methylenetetrahydrofolate reductase (MTHFR) polymorphisms. These genetic alterations lead to a hypercoagulable state, which can compromise placental perfusion and function through mechanisms such as microvascular thrombosis, trophoblastic dysfunction, and altered angiogenesis [6,7,8,9,10].

The adverse pregnancy outcomes associated with inherited thrombophilias are well-documented. Women with thrombophilia are at significantly increased risk for complications such as recurrent pregnancy loss, preeclampsia, intrauterine growth restriction (IUGR), and stillbirth. These complications are largely attributed to impaired placental development and function, driven by thrombosis and inflammation. Furthermore, disruptions in placental health can alter the levels of biomarkers used in first-trimester screening, potentially impacting the accuracy of risk assessments [11,12].

Reduced levels of PAPP-A, a marker produced by the placenta, are frequently observed in pregnancies affected by thrombophilias. Low PAPP-A levels are associated with poor placental perfusion and trophoblastic dysfunction, which correlate with increased risks of adverse outcomes, including preeclampsia and fetal growth restriction. Similarly, aberrations in β-hCG levels, which reflect trophoblastic activity, have been reported in pregnancies complicated by thrombophilias. These alterations may contribute to the misclassification of chromosomal abnormality risk in first-trimester screening [13,14,15].

Ultrasonographic parameters such as NT, while primarily used to assess fetal chromosomal anomalies, may also be indirectly influenced by placental insufficiency in thrombophilic pregnancies. Although NT is not directly linked to thrombophilias, the systemic effects of these conditions, such as reduced placental function, could lead to subtle alterations in NT measurements, potentially affecting risk calculations in first-trimester screening [16,17].

The clinical implications of thrombophilias on first-trimester screening are significant. Altered biomarker levels and indirect effects on ultrasonographic parameters can result in false-positive or false-negative outcomes in risk assessments, increasing maternal anxiety or leading to unnecessary interventions. For example, underestimating the role of thrombophilias in low PAPP-A levels may lead to inappropriate conclusions about fetal chromosomal anomalies. Conversely, failing to recognize the contribution of thrombophilias to placental insufficiency could delay the implementation of preventive measures [18,19].

Current evidence highlights the importance of identifying and managing thrombophilias in pregnant women to optimize screening outcomes. A meta-analysis by Rodger et al. (2014) emphasized the need for tailored approaches to screening in women with known thrombophilias, advocating for the adjustment of biomarker interpretation to account for the effects of these conditions. Additionally, early identification of thrombophilic mutations through genetic screening can guide the use of anticoagulant therapies, such as low-molecular-weight heparin, to reduce the risk of placental thrombosis and improve pregnancy outcomes [19,20].

This paper aims to provide a comprehensive analysis of the impact of inherited thrombophilic mutations on first-trimester screening. By integrating evidence from prospective cohort studies, systematic reviews, and meta-analyses, it seeks to elucidate the mechanisms through which thrombophilias affect placental biomarkers and screening outcomes. Furthermore, it will explore strategies to refine screening protocols for women with thrombophilias, enhancing the predictive accuracy of these tests and improving maternal–fetal outcomes.

## 2. Materials and Methods

### 2.1. Study Design

This observational descriptive study was conducted to evaluate the impact of inherited thrombophilic mutations on first-trimester screening outcomes in pregnant women. The study was carried out at “Spitalul Clinic Județean de Urgență Bihor” between September 2020 and September 2024. Ethical approval was granted by the Institutional Review Board, and all participants provided written informed consent prior to enrollment.

### 2.2. Study Population

A total of 105 pregnant women were recruited from the “Spitalul Clinic Județean de Urgență Bihor”, during their first trimester (10–13 weeks of gestation). The cohort was divided into two groups: group I (1), consisting of patients with thrombophilia; and group II (2), consisting of patients without thrombophilia. The inclusion criteria were singleton pregnancies confirmed by ultrasound, maternal age ≥ 18 years, and absence of chronic conditions unrelated to thrombophilia (e.g., diabetes, hypertension). The exclusion criteria included multiple gestations, maternal history of autoimmune disorders or anticoagulant therapy unrelated to thrombophilia, and chromosomal abnormalities diagnosed prenatally, as patients with systemic lupus erythematosus (SLE), antiphospholipid syndrome (APS), rheumatoid arthritis, and other documented autoimmune disorders that may impact pregnancy outcomes or thrombophilia risk assessment.

### 2.3. Assessment of Thrombophilic Mutations

Peripheral blood samples (5 mL) were collected from all participants at the time of enrollment for genetic analysis. DNA extraction was performed using silica-based column kits (e.g., QIAamp DNA Blood Mini Kit, Qiagen, Cluj-Napoca, Romania) or automated systems to ensure high-quality genomic DNA. Genotyping was performed using real-time polymerase chain reaction (PCR) with specific primers and probes, and internal controls were included in the assays to ensure accuracy. The following thrombophilic mutations were analyzed in our study: factor V Leiden (G1691A) mutation, prothrombin gene (G20210A) mutation, methylenetetrahydrofolate reductase (MTHFR) polymorphisms (C677T and A1298C), protein C deficiency, protein S deficiency, antithrombin III deficiency, factor XIII mutation, plasminogen activator inhibitor-1 (PAI-1) polymorphism, and ACE I/D polymorphism. Their respective frequencies are now detailed in the manuscript. Based on the results of these analyses, participants underwent a comprehensive evaluation by a hematologist, who confirmed the diagnosis of thrombophilia in accordance with established clinical and laboratory criteria.

### 2.4. First-Trimester Screening

All participants underwent standard first-trimester screening at 11–13 weeks of gestation. Biochemical marker measurements included free β-hCG and PAPP-A levels, analyzed using the BRAHMS Kryptor System (Thermo Fisher Scientific, Bangkok, Thailand), which employs the time-resolved amplified cryptate emission (TRACE) technology to ensure high precision and reproducibility. Results were expressed as multiples of the median (MoM), adjusted for maternal age, weight, and gestational age, and corrected using the FMF risk calculation (Brahms Kryptor) in accordance with the guidelines set by the Fetal Medicine Foundation. Ultrasonographic evaluation was conducted using a Voluson E10 Expert BTXX (Zipf, Austria), with certified sonographers performing the measurements according to the Fetal Medicine Foundation guidelines.

### 2.5. Data Collection

Maternal demographic and clinical data, including age, parity, body mass index (BMI), and history of pregnancy complications, and the risk factors (smoking, chronic hypertension, previous pregnancy-induces hypertension, small-for-gestational-age infant) were recorded. The primary outcomes of interest included levels of biochemical markers (free β-hCG and PAPP-A), NT measurements, and pregnancy complications such as preeclampsia, intrauterine growth restriction (IUGR), and preterm birth.

### 2.6. Statistical Analysis

Data were analyzed using SPSS (version 20). Descriptive statistics were used to summarize the data, with results presented as mean ± standard deviation (SD) or median with interquartile range (IQR) for continuous variables, and as frequencies and percentages for categorical variables. Group comparisons, such as thrombophilia-positive versus thrombophilia-negative, were performed using independent *t*-tests or Mann–Whitney U-tests for continuous variables and chi-square tests or Fisher’s exact tests for categorical variables. A multivariate regression model was applied to evaluate the independent effects of thrombophilic mutations on first-trimester screening parameters, adjusting for confounders such as maternal age, BMI, and smoking status. Statistical significance was set at *p* < 0.05.

## 3. Results

### 3.1. Demographic Description

The study population had a mean maternal age of 32.3 years (SD ± 4.4), with ages ranging from 23 to 41 years. The mean maternal weight was 67.7 kg (SD ± 12.93), ranging from 48 to 114 kg, and the average height was 165.21 cm (SD ± 5.94), ranging from 150 to 183 cm. The mean body mass index (BMI) was 24.77 (SD ± 4.39), with a minimum of 17.71 and a maximum of 42.18. The participants were, on average, 12.29 weeks pregnant (SD ± 0.76), with a minimum gestational age of 11 weeks and a maximum of 14 weeks. Additionally, the average parity was 0.84 (SD ± 1.04), ranging from 0 to 5. The descriptive statistics of maternal and pregnancy characteristics in each study group and at cohort level are presented in Table 1.

### 3.2. Risk Factors

Figure 1 summarizes the health- and pregnancy-related characteristics of the study participants. Most participants (77.1%) were non-smokers, while 22.9% reported being smokers. Chronic hypertension was rare, affecting only 1% of the participants, and none had lupus or antiphospholipid syndrome. Previous pregnancy-induced hypertension occurred in 2.9% of participants, while 97.1% had no such complications. Additionally, 1.9% of participants reported a history of having a small-for-gestational-age infant, with 98.1% having no such history. Furthermore, none of the participants reported a maternal history of hypertension (as heredocollateral antecedents), indicating the overall low prevalence of chronic health conditions within the study population.

Figure 1 compares risk factors between the two study groups. In Group 1, 28.6% of participants were non-smokers, and 12.4% were smokers; while in Group 2, 48.6% were non-smokers, and 10.5% were smokers. None of the participants in Group 1 had chronic hypertension, while 41% were free of it. In Group 2, 58.1% of participants did not have chronic hypertension, and 1% were affected. Regarding pregnancy-induced hypertension, none of the participants in Group 1 experienced it, with 41% remaining unaffected; whereas in Group 2, 56.2% did not have pregnancy-induced hypertension, and 2.9% experienced it. Similarly, none of the participants in Group 1 had a history of small-for-gestational-age infants, while 41% were free of this history. In Group 2, 57.1% of participants did not have this history, while 1.9% reported having it. Overall, the figure highlights slight differences in risk factors between the two groups, with Group 2 showing a slightly higher prevalence of chronic and pregnancy-related conditions. Risk factors for each study group are presented in Figure 1.

### 3.3. Ultrasound Parameters

Figure 2 presents the descriptive statistics for first-trimester screening parameters. The mean crown–rump length (CRL) was 64.74 mm (SD ± 9.56), and the average nuchal translucency (NT) thickness was 1.53 mm (SD ± 0.56). The mean free β-hCG level was 50.22 IU/L (SD ± 31.72), with an adjusted MoM of 1.27 (SD ± 0.72). PAPP-A levels averaged 3.64 IU/L (SD ± 2.52), with an adjusted MoM of 1.17 (SD ± 0.68). These values highlight the variability in fetal measurements and biochemical markers within the study population.

Figure 2 compares first-trimester screening parameters between two study groups. In Group 1, the mean crown–rump length (CRL) was 59.41 mm (SD ± 8.53), compared to 68.44 mm (SD ± 8.47) in Group 2. The mean nuchal translucency (NT) thickness was 1.09 mm (SD ± 0.38) in Group 1 and 1.83 mm (SD ± 0.45) in Group 2. Free β-hCG levels averaged 47.89 IU/L (SD ± 32.30) in Group 1 and 51.83 IU/L (SD ± 31.48) in Group 2, with similar MoM values of 1.29 (SD ± 0.74) and 1.25 (SD ± 0.71), respectively. PAPP-A levels were higher in Group 1 (4.21 IU/L; SD ± 2.70) than in Group 2 (3.25 IU/L; SD ± 2.32), although adjusted MoM values were comparable at 1.12 (SD ± 0.71) and 1.20 (SD ± 0.66). These results highlight differences in fetal measurements and placental marker levels between the groups (Figure 2).

### 3.4. Correlation

Table 2 presents the correlations between various maternal, fetal, and clinical parameters, with statistically significant relationships highlighted. For crown–rump length (CRL or LCC), there is a weak positive correlation with maternal age (r = 0.240; *p* = 0.014), indicating that CRL slightly increases with older maternal age. A strong positive correlation with gestational weeks (r = 0.867; *p* = 0.000) reflects the expected growth of the fetus as pregnancy progresses. Conversely, a weak negative correlation with parity (r = −0.197; *p* = 0.044) suggests that higher parity is associated with slightly smaller CRL measurements.

For nuchal translucency (NT), a weak positive correlation with maternal age (r = 0.210, *p* = 0.031) and maternal weight (r = 0.193; *p* = 0.049) suggests that NT thickness tends to increase slightly in older or heavier mothers. Additionally, a moderate positive correlation with gestational weeks (r = 0.471; *p* = 0.000) indicates that NT thickness grows as pregnancy advances.

Regarding free β-hCG (IU/L), there is a weak negative correlation with maternal weight (r = −0.264, *p* = 0.007) and a moderate negative correlation with BMI (r = −0.329; *p* = 0.001), indicating that higher maternal weight and BMI are associated with lower free β-hCG levels. A weak negative correlation with gestational weeks (r = −0.216; *p* = 0.027) further suggests that free β-hCG levels decrease slightly as pregnancy progresses.

For PAPP-A, weak negative correlations are observed with maternal weight (r = −0.249; *p* = 0.010) and BMI (r = −0.247; *p* = 0.011), indicating that higher maternal weight and BMI are linked to lower PAPP-A levels. Additionally, PAPP-A MoM shows a weak positive correlation with maternal age (r = 0.209; *p* = 0.032), suggesting slightly higher adjusted PAPP-A values in older mothers.

Overall, gestational age strongly correlates with CRL and moderately with NT, reflecting normal fetal growth patterns during the first trimester. Maternal weight and BMI negatively affect free β-hCG and PAPP-A levels, highlighting their influence on biomarker concentrations. Weak positive correlations with maternal age are observed across several parameters, while parity shows a weak negative correlation with CRL, indicating slightly smaller fetal size in mothers with higher parity. These findings provide critical insights into the interactions between maternal and fetal characteristics during first-trimester screening.

Table 3 presents the Pearson correlation analysis among first-trimester screening parameters, highlighting statistically significant relationships. Crown–rump length (CRL or LCC) shows a strong positive correlation with nuchal translucency (NT) thickness (r = 0.551; *p* = 0.000), indicating that as fetal size increases, NT thickness also tends to increase. Additionally, there is a weak positive correlation between CRL and PAPP-A levels (r = 0.232, *p* = 0.017), suggesting slightly higher placental protein levels in larger fetuses. No other significant correlations were observed for CRL.

Nuchal translucency (NT) is negatively correlated with free β-hCG levels (r = −0.238; *p* = 0.014) and free β-hCG MoM (r = −0.255; *p* = 0.009). These findings suggest that higher NT thickness is associated with lower levels of free β-hCG and its MoM values (excluding Edwards and Patau signs). This may reflect an interaction between fetal development and trophoblastic activity.

For free β-hCG, there is a strong positive correlation with free β-hCG MoM (r = 0.902; *p* = 0.000), confirming the expected relationship between raw biomarker values and their normalized MoM. Free β-hCG also shows a weak positive correlation with PAPP-A MoM (r = 0.231; *p* = 0.018), indicating some alignment between these biomarkers when adjusted for maternal and gestational factors.

Free β-hCG MoM demonstrates a positive correlation with PAPP-A (r = 0.259; *p* = 0.008) and PAPP-A MoM (r = 0.298; *p* = 0.002), indicating that higher free β-hCG MoM levels are associated with increased placental protein levels. This finding highlights the interdependence of these biomarkers in representing placental function.

Additionally, PAPP-A and PAPP-A MoM exhibit a strong positive correlation (r = 0.789; *p* < 0.001), which is expected due to their inherent relationship. This consistency reflects the alignment between raw and normalized values of this placental biomarker.

In summary, the strongest correlations observed were between related measures (e.g., free β-hCG and its MoM; PAPP-A and its MoM), while weaker but significant correlations highlighted interactions between fetal growth (CRL and NT) and biomarker levels. These findings emphasize the interconnection between fetal size, placental function, and first-trimester screening markers.

### 3.5. Relative Risk

The risk estimate figure provides insights into the association between smoking status and low PAPP-A levels (<1.0 MoM), presented in Figure 3. The odds ratio (OR) for smoking (non-smokers vs. smokers) is 0.400, with a 95% confidence interval (CI) ranging from 0.103 to 1.555. This suggests that non-smokers may have a reduced likelihood of low PAPP-A compared to smokers; however, the wide confidence interval crossing 1 indicates that this result is not statistically significant. For the cohort with low PAPP-A (<1.0 MoM), the OR is 0.444 (95% CI: 0.137–1.447), indicating a potential protective trend, but it is also not statistically significant. In the cohort with normal or high PAPP-A (≥1.0 MoM), the OR is 1.111 (95% CI: 0.920–1.343), suggesting a slight increase in odds; but again, the confidence interval crossing 1 means the association is not significant. With 105 valid cases analyzed, these results imply no clear or statistically significant relationship between smoking status and PAPP-A levels in this dataset.

The risk estimate figure evaluates the association for the cohort with nuchal translucency (NT) < 2.5 mm. The odds ratio (OR) is 0.962, with a 95% confidence interval (CI) of 0.925 to 0.999. This suggests a very slight protective effect for having NT < 2.5 mm, as the OR is below 1. However, the confidence interval is very close to 1, indicating that the association is marginal and likely not clinically significant. With 105 valid cases, this result implies no strong or statistically significant relationship between the evaluated factors and NT measurements below 2.5 mm.

The risk estimate evaluates the association for the cohort where participants reported no history of having a small-for-gestational-age infant (copil mic anterior = nu). The odds ratio (OR) is 3.000, with a 95% confidence interval (CI) ranging from 0.606 to 14.864. This indicates that participants without a history of small-for-gestational-age infants may have a threefold higher likelihood of the outcome being studied compared to those with such a history. However, the wide confidence interval, which includes 1, suggests that this result is not statistically significant. With 105 valid cases analyzed, this finding suggests a possible trend but lacks sufficient statistical evidence to confirm a strong or reliable association.

The figure evaluates the association between parity and the likelihood of low PAPP-A levels (<1.0 MoM). The odds ratio (OR) for parity (≤2 vs. >2) is 0.173, with a 95% confidence interval (CI) of 0.044 to 0.676, indicating that women with parity ≤2 are significantly less likely to have low PAPP-A levels compared to those with parity >2. For the cohort with low PAPP-A (<1.0 MoM), the OR is 0.221 (95% CI: 0.071–0.688), further supporting a reduced likelihood of low PAPP-A in women with parity ≤2. However, in the cohort with normal or high PAPP-A (≥1.0 MoM), the OR is 1.278 (95% CI: 0.972–1.681), suggesting a slight but statistically insignificant increase in odds for higher parity. With 105 valid cases, these results highlight a significant protective effect of lower parity (≤2) against low PAPP-A levels.

### 3.6. Multivariate Linear Regression

The multivariate regression models assess how maternal factors such as smoking, chronic hypertension, pregnancy-induced hypertension, and parity impact first-trimester screening markers, specifically PAPP-A levels and nuchal translucency (NT).

For PAPP-A, smoking showed a consistent negative association but was not statistically significant (*p* > 0.05). Chronic and pregnancy-induced hypertension had minimal, non-significant effects. However, parity had a significant negative impact (B = −1.428; *p* = 0.026), indicating that higher parity is linked to lower PAPP-A levels. This suggests parity as the strongest predictor of PAPP-A levels, while other maternal factors had negligible effects.

For NT, smoking had a weak negative association, while chronic and pregnancy-induced hypertension showed small positive effects; none reached statistical significance. Parity showed a moderate negative trend (B = −0.250; *p* = 0.079), nearing significance but not conclusive. Overall, no independent variable significantly predicted NT, with parity showing the most notable, albeit non-significant, effect (Table 4).

Parity is the most significant predictor of PAPP-A levels, with higher parity being linked to lower PAPP-A (*p* = 0.026). In contrast, smoking and hypertension, both chronic and pregnancy-induced, do not have a significant effect on either PAPP-A or NT. While none of the predictors show a strong association with NT thickness, parity exhibits a slight negative trend, though it does not reach statistical significance.

Figure 4 illustrates the multivariate regression coefficients for PAPP-A and NT, highlighting the effects of smoking, chronic hypertension, pregnancy-induced hypertension, and parity. Parity emerges as the strongest predictor, showing a significant negative association with PAPP-A levels (*p* = 0.026), while its effect on NT is weaker and only borderline significant (*p* = 0.079). Smoking and hypertension (both chronic and pregnancy-related) exhibit minimal influence on PAPP-A and NT, with no statistically significant impact. The figure visually demonstrates these relationships, emphasizing parity as the most influential maternal factor affecting first-trimester screening markers.

## 4. Discussion

This prospective study, conducted between 2020 and 2024, focused on a cohort of pregnant women during their first trimester in the Romania. Thrombophilia has been extensively associated with maternal and fetal complications, raising the hypothesis that disruptions in maternal hemostasis may adversely affect placental function, ultimately leading to impaired fetal growth and development [21].

Potential mechanisms contributing to increased nuchal translucency (NT) include cardiac failure linked to abnormalities of the heart and great arteries, venous congestion in the head and neck caused by factors such as fetal body constriction in amnion rupture sequence or superior mediastinal compression associated with conditions like diaphragmatic hernia or a narrow chest in skeletal dysplasia, and alterations in the composition of the extracellular matrix, potentially resulting from gene dosage effects [22,23]. Our study identified a significant association between lower parity (≤2) and a reduced risk of low PAPP-A levels (<1.0 MoM), suggesting that lower parity may serve as a protective factor in placental function. While non-smokers showed a trend toward reduced risk of low PAPP-A, the association was not statistically significant. Nuchal translucency (NT) measurements below 2.5 mm were largely consistent with normal fetal development, and maternal factors such as chronic hypertension and a history of small-for-gestational-age infants did not exhibit statistically significant correlations with first-trimester screening markers.

The protective effect of lower parity on low PAPP-A levels aligns with findings from previous studies, which have reported that higher parity is associated with impaired placental function and increased risks of adverse pregnancy outcomes. This relationship may be explained by cumulative vascular damage caused by multiple pregnancies, as highlighted in the literature. These findings emphasize the need to account for parity in first-trimester screening to improve the interpretation of biomarkers like PAPP-A, especially in multiparous women [24,25,26]. It is important to note that once all relevant data, including weight, parity, smoking status, β-hCG, and PAPP-A, are entered into the T1 software PRISCA (PerkinElmer) and LifeCycle (Revvity), the MoM values for β-hCG and PAPP-A are adjusted accordingly, ensuring that factors such as smoking, obesity, or previous pregnancies do not influence the corrected MoM values.

In smokers, the detection rate of trisomy 21 using free β-hCG, PAPP-A, and maternal age is estimated to be reduced by approximately 5–6% compared to the general population [27,28]. Although smoking was not significantly associated with low PAPP-A levels in our cohort, previous research has established smoking as a risk factor for placental dysfunction. The absence of significance in our study may be due to the relatively small sample size or variability in smoking intensity and duration among participants. Nonetheless, the observed trend underscores the importance of smoking cessation programs as part of prenatal care to optimize placental health and pregnancy outcomes.

In a study from 2015, it was observed that the reference range identified in the studied population differed significantly from those reported in other ethnic groups. This finding suggested that using population-specific values may enhance the effectiveness of first-trimester screening for chromosomal abnormalities compared to relying on a single universal cutoff value [29,30,31]. The observation that NT measurements < 2.5 mm align with normal fetal development is consistent with findings in other studies, which have similarly defined this range as normal. However, elevated NT (>3.5 mm) remains a strong predictor of chromosomal abnormalities, as emphasized in systematic reviews. This highlights the ongoing importance of NT as a key component of first-trimester screening, while further investigation is warranted to refine thresholds for improved predictive accuracy.

These findings underline the value of incorporating maternal factors, such as parity and smoking status, into first-trimester risk assessments to enhance screening accuracy. Adjusting biomarker interpretation for these factors may provide a more individualized assessment of pregnancy risk. The results also suggest that targeted interventions, such as smoking cessation programs, could positively impact placental function and pregnancy outcomes.

One limitation of this study is the relatively small sample size, which may reduce the power to detect significant associations for certain factors, such as smoking. Additionally, confounding variables, including maternal stress or comorbidities, were not fully accounted for, which may influence biomarker levels. However, the prospective design of the study and adherence to standardized screening protocols strengthen the reliability of the findings. The relatively small sample size limits the statistical power and generalizability of our findings. Additionally, the study was conducted in a single-center setting, which may introduce selection bias and limit the external validity of our results. Future studies should incorporate larger cohorts and multiple institutions to ensure broader applicability and improved statistical robustness. This study does not include follow-up data on pregnancy outcomes, limiting its ability to establish causality between thrombophilia and first-trimester screening markers. Future research should incorporate longitudinal follow-up to assess clinical outcomes and validate our findings. This study does not include certain additional placental biomarkers (PlGF, sFlt-1) and uterine artery Doppler studies, which could have provided a more comprehensive assessment. These were excluded due to incomplete or inconsistent data. Future studies should aim to incorporate these markers while ensuring data quality. Although efforts were made to control for confounding variables, residual confounding cannot be entirely ruled out. Further studies with larger, more diverse populations and robust statistical adjustments are necessary to strengthen causal inferences.

Future research with larger and more diverse populations is needed to confirm the protective role of lower parity on placental biomarkers and to explore the underlying mechanisms. Longitudinal studies could also examine the cumulative effects of maternal risk factors, such as smoking and parity, on pregnancy outcomes and refine screening thresholds for biomarkers like PAPP-A and NT to improve clinical utility.

## 5. Conclusions

This study investigates the relationship between inherited thrombophilias and modifications in first-trimester prenatal screening markers, including PAPP-A, β-hCG, and NT. Our findings suggest that thrombophilia-related alterations in placental function influence these biomarkers, potentially affecting the accuracy of prenatal risk assessments. Women with thrombophilia exhibited distinct variations in first-trimester screening parameters, emphasizing the need for tailored risk interpretation in this population. Given the potential implications for early pregnancy management, incorporating thrombophilia screening into first-trimester assessments may enhance prenatal care strategies. However, due to the limitations of our study, including the single-center design and relatively small sample size, further multi-center and longitudinal research is necessary to validate these findings and refine clinical recommendations.

## Figures and Tables

**Figure 1 medicina-61-00318-f001:**
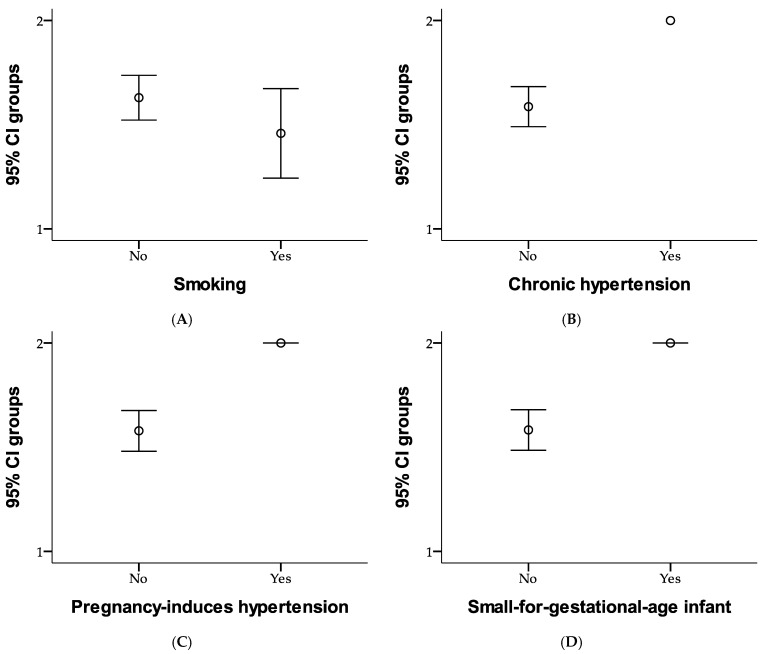
Comparison of risk factors as smoking (**A**), chronic hypertension (**B**), pregnancy-induced hypertension (**C**), and small-for-gestational-age infant (**D**) between study groups using error bar method.

**Figure 2 medicina-61-00318-f002:**
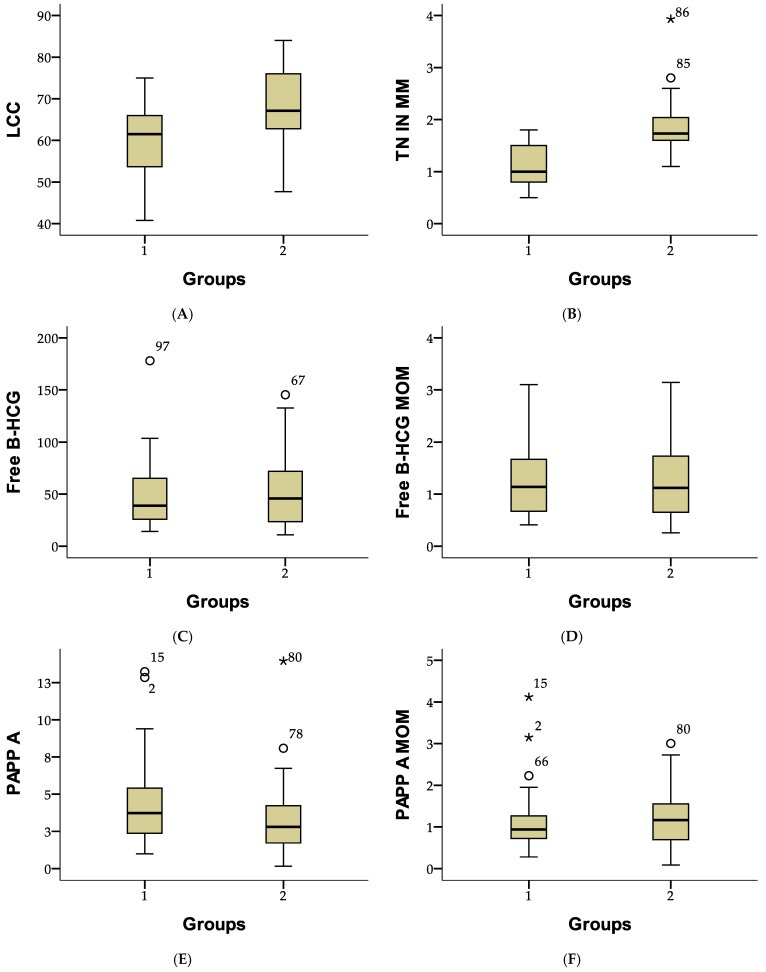
Comparison of fetal and biochemical parameters between study groups using boxplot representation. (**A**) Crown-rump length (LCC), (**B**) Nuchal translucency (TN), (**C**) Free β-hCG, (**D**) Free β-hCG MoM, (**E**) Pregnancy-associated plasma protein-A (PAPP-A), and (**F**) PAPP-A MoM. The central box represents the interquartile range (IQR), with the horizontal line indicating the median. Whiskers extend to the minimum and maximum values within 1.5 times the IQR. White circles (○) indicate mild outliers, while stars (*) represent extreme outliers.

**Figure 3 medicina-61-00318-f003:**
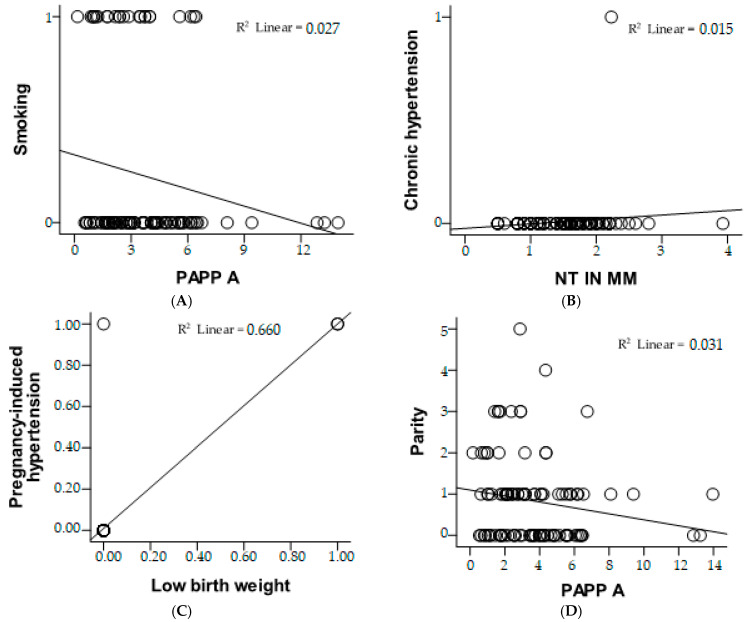
Correlation analysis between clinical and biochemical parameters using scatter plots with linear regression lines. (**A**) Relationship between PAPP-A and smoking status, showing a weak negative correlation (*R*^2^ = 0.027). (**B**) Association between NT measurement and chronic hypertension, with a very low correlation (*R*^2^ = 0.015). (**C**) Strong correlation between pregnancy-induced hypertension and low birth weight (*R*^2^ = 0.660), suggesting a significant association. (**D**) Relationship between PAPP-A and parity, indicating a weak negative correlation (*R*^2^ = 0.031).

**Figure 4 medicina-61-00318-f004:**
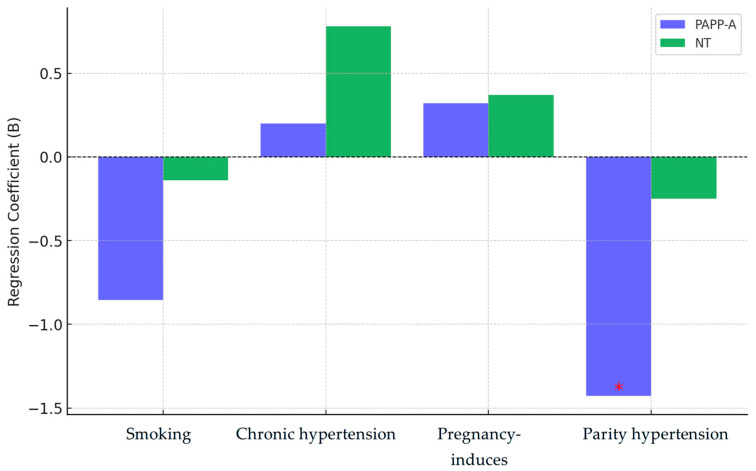
Multivariate regression coefficients for PAPP-A and NT. The red asterisk (*) denotes a statistically significant value (e.g., *p* < 0.05), highlighting a significant effect in the regression analysis.

**Table 1 medicina-61-00318-t001:** Descriptive statistics of maternal and pregnancy characteristics in each study group and at cohort level.

Parameters	Groups				Total	
	I		II			
	Mean	SD	Mean	SD	Mean	SD
Age	29.84	3.97	34.00	3.87	32.30	4.40
Maternal weight	68.74	13.62	66.98	12.49	67.70	12.93
Height	163.79	5.03	166.19	6.36	165.21	5.94
BMI	25.60	4.91	24.20	3.92	24.77	4.39
Weeks pregnant	11.95	0.65	12.52	0.74	12.29	0.76
Days of pregnancy accuracy	3.35	2.06	3.37	2.13	3.36	2.09
Parity	0.98	1.26	0.74	0.85	0.84	1.04

SD = standard deviation; I = group consisting of patients with thrombophilia; II = group consisting of patients without thrombophilia.

**Table 2 medicina-61-00318-t002:** Pearson correlation analysis between maternal, fetal, and first-trimester screening parameters.

Pearson Correlation		Age	Maternal Weight	Height	BMI	Weeks Pregnant	Days of Pregnancy
LCC	r	0.240 *	−0.021	0.046	−0.047	0.867 **	0.104
	*p*	0.014	0.834	0.641	0.632	0.000	0.291
TN IN MM	r	0.210 *	0.042	0.193 *	−0.042	0.471 **	0.036
	*p*	0.031	0.671	0.049	0.673	0.000	0.719
Free B HCG	r	0.034	−0.264 **	0.099	−0.329 **	−0.216 *	−0.020
	*p*	0.729	0.007	0.313	0.001	0.027	0.840
Free B HCG MOM	r	0.050	−0.093	0.166	−0.176	−0.075	−0.044
	*p*	0.612	0.344	0.090	0.073	0.450	0.655
PAPP A	r	0.059	−0.249 *	−0.047	−0.247 *	0.167	0.130
	*p*	0.550	0.010	0.635	0.011	0.090	0.186
PAPP A MOM	r	0.209 *	−0.048	0.111	−0.096	0.081	0.003
	*p*	0.032	0.630	0.261	0.329	0.411	0.977
N		105					

N = number of patients; r = Pearson coefficient; *p* = statistically significance; ** = correlation is significant at the 0.01 level (two-tailed); * = correlation is significant at the 0.05 level (two-tailed).

**Table 3 medicina-61-00318-t003:** Pearson correlation analysis among first-trimester screening parameters.

Pearson Correlation		LCC	TN IN MM	Free B HCG	Free B HCG MoM	PAPP A	PAPP A MoM
LCC	r	1	0.551 **	−0.182	−0.034	0.232 *	0.115
	*p*	-	0.000	0.064	0.728	0.017	0.243
TN IN MM	r	0.551 **	1	−0.238 *	−0.255 **	−0.036	0.058
	*p*	0.000	-	0.014	0.009	0.716	0.556
Free B HCG	r	−0.182	−0.238 *	1	0.902 **	0.161	0.231 *
	*p*	0.064	0.014	-	0.000	0.101	0.018
Free B HCG MoM	r	−0.034	−0.255 **	0.902 **	1	0.259 **	0.298 **
	*p*	0.728	0.009	0.000	-	0.008	0.002
PAPP A	r	0.232 *	−0.036	0.161	0.259 **	1	0.789 **
	*p*	0.017	0.716	0.101	0.008	-	0.000
PAPP A MoM	r	0.115	0.058	0.231 *	0.298 **	0.789 **	1
	*p*	0.243	0.556	0.018	0.002	0.000	-
	N	105	105	105	105	105	105

N = number of patients; r = Pearson coefficient; *p* = statistical significance; ** = correlation is significant at the 0.01 level (two-tailed); * = correlation is significant at the 0.05 level (two-tailed).

**Table 4 medicina-61-00318-t004:** Impact of maternal factors on first-trimester screening markers (PAPP-A and NT).

Predictor	Effect on PAPP-A	Significance	Effect on NT	Significance	Predictor
Smoking	Negative	No	Negative	No	Smoking
Chronic Hypertension	Positive	No	Positive	No	Chronic Hypertension
Pregnancy-Induced Hypertension	Minimal	No	Minimal	No	Pregnancy-Induced Hypertension
Parity	Negative	Yes (*p* = 0.026)	Negative	Borderline (*p* = 0.079)	Parity

## Data Availability

All the data processed in this article are part of the research for a doctoral thesis, being archived in the aesthetic medical office where the interventions were performed.

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
