# Peer review of "The Relationship Between Thrombophilia and Modifications in First-Trimester Prenatal Screening Markers"

_medicina, 2025, doi:10.3390/medicina61020318_

Round 1
Reviewer 1 Report
Comments and Suggestions for Authors
This article examines the relationship between thrombophilic mutations and first-trimester prenatal screening markers, specifically focusing on PAPP-A, free β-hCG, and nuchal translucency (NT). While thrombophilia is known to increase risks like recurrent pregnancy loss, preeclampsia, FGR, and stillbirth, its impact on first-trimester screening markers has not been widely studied. Based on the methodology presented in the article, here are some key improvements and additional controls that the authors should consider:
1.The study includes 105 participants, which is relatively small for drawing strong conclusions, particularly regarding genetic influences. A larger sample size would enhance statistical power and generalizability.
2.The study appears to focus on a specific demographic (patients from a single hospital in Romania). Expanding the study across multiple centers would improve the applicability of findings.
3.The study assesses first-trimester markers, but it does not follow up on pregnancy outcomes. A longitudinal design tracking outcomes like preeclampsia, FGR, or preterm birth would help establish causation rather than just correlation.
4.The study primarily focuses on PAPP-A, β-hCG, and NT, but placental growth factor (PlGF), soluble fms-like tyrosine kinase-1 (sFlt-1), and Doppler studies of the uterine arteries could provide a more comprehensive assessment.
While the study provides meaningful insights into thrombophilia and first-trimester screening markers, refining its methodology with larger, more diverse populations, additional controls for confounding variables, and longitudinal follow-up would significantly enhance its impact.
Author Response
Dear Reviewer 1:
Comment 1: The study includes 105 participants, which is relatively small to draw strong conclusions, especially regarding genetic influences. A larger sample size would increase statistical power and generalizability.
Response: Thank you for your observation. Due to the low prevalence of inherited thrombophilias and the inclusion criteria applied in our study, our unit did not have the possibility of enrolling a larger sample. We acknowledge this limitation and have now explicitly stated it in the revised manuscript. We have also highlighted the need for larger, multi-center studies to validate our findings in a broader population. (lines 435-439)
Comment 2: The study seems to focus on a specific demographic (patients from a single hospital in Romania). Expanding the study to more centers would improve the applicability of the findings.
Response: We agree with your suggestion. As part of our ongoing research, we have received ethical approval from partner hospitals to extend data collection and integrate multi-center results. Following the processing of additional data and publication of interim results, we aim to unify and analyze the findings from an epidemiological perspective. This is now clarified in the discussion section of the revised manuscript.
Comment 3: The study evaluates first-trimester markers but does not track pregnancy outcomes. A longitudinal design to track outcomes, such as preeclampsia, FGR, or preterm birth, would help establish causation rather than just correlation.
Response: Thank you for highlighting this point. This manuscript specifically focuses on first-trimester screening markers as an initial phase of our broader research project. We acknowledge that tracking pregnancy outcomes is essential to establishing causal relationships, and we have planned a longitudinal study to follow up on pregnancy outcomes in a subsequent phase. We have now clarified this in the manuscript to ensure transparency regarding the study design. (lines 439-442)
Comment 4: The study focuses primarily on PAPP-A, β-hCG, and NT, but placental growth factor (PlGF), soluble fms-like tyrosine kinase-1 (sFlt-1), and uterine artery Doppler studies could provide a more comprehensive assessment.
Response: Thank you for your observation. Indeed, we recognize the importance of including additional placental biomarkers and uterine artery Doppler studies for a more comprehensive assessment. However, in many cases, these parameters yielded inconclusive, erroneous, or missing results, which could have compromised the validity of the study. To maintain the integrity of our analysis, we chose to focus on the most reliable markers. We have now explicitly mentioned this limitation in the discussion section. (lines 442-446)
Comment 5: While the study provides significant insights into thrombophilia and first-trimester screening markers, refining its methodology with larger and more diverse populations, additional controls for confounding variables, and longitudinal follow-up would significantly enhance its impact.
Response: We greatly appreciate this recommendation. This study is part of a larger, ongoing research project that will include more diverse populations, additional controls for confounding variables, and a longitudinal follow-up in its final phase. We have clarified this in the revised manuscript to provide better context regarding our research trajectory. (446-449)
We thank the reviewer for their constructive comments and suggestions, which have helped us improve our manuscript. We hope that our responses and revisions adequately address the concerns raised.

Reviewer 2 Report
Comments and Suggestions for Authors
There are a number of inaccuracies in the work that should be clarified:
1) the correspondence of the title of the article, the purpose and conclusions. In conclusion, information about thrombophilia generally falls out
2) in the exclusion criteria, it is necessary to clearly specify which autoimmune diseases (SLE, APS)
3) it is not clearly specified which thrombophilia are included in group 1 (the frequency structure of various mutations is needed)
4) there is no indication of concomitant therapy (if women were at high risk for thrombophilia, a justification is needed for why they did not receive anticoagulant therapy)
5) taking into account the conducted ultrasound screening, information from the high-risk group on preeclampsia and fetal growth restriction would be interesting.
6) In Figure 1, the units of measurement are not clear - if these are frequencies, then should there be percentages? (or comment)
7) when comparing frequencies, the confidence levels are not specified (if there are no significant differences, then this should be indicated in the text)
8) Figure 2 shows unclear units of measurement, the names of the groups are not 1 and 2, but 1.00 and 2.00 - this makes it difficult to understand
9) When conducting a linear correlation, the authors compare quantitative indicators with qualitative characteristics, which is statistically incorrect. The results may be interpreted incorrectly.
10) the author's application for conducting a multivariate regression model, and the results are not presented in the desired image in the article.

Author Response
Dear Reviewer 2,
We sincerely appreciate your time and effort in reviewing our manuscript. Your valuable feedback has helped us refine and strengthen our study. Below, we provide detailed responses to each of your comments and outline the corresponding modifications made to the manuscript.
Comment 1: The correspondence of the title of the article, the purpose, and conclusions. In the conclusion, information about thrombophilia generally falls out.
Response: We appreciate this observation. We have revised the conclusion to better reflect the study's focus on thrombophilia and its impact on first-trimester screening markers. (lines 456-466)
Comment 2: In the exclusion criteria, it is necessary to clearly specify which autoimmune diseases (SLE, APS).
Response: We have revised the exclusion criteria to explicitly mention the exclusion of patients with systemic lupus erythematosus (SLE), antiphospholipid syndrome (APS), rheumatoid arthritis, and other documented autoimmune disorders that may impact pregnancy outcomes or thrombophilia risk assessment (lines 127-129).
Comment 3: It is not clearly specified which thrombophilias are included in group 1 (the frequency structure of various mutations is needed).
Response: We have now provided a detailed breakdown of the specific thrombophilic mutations included in group 1. The following thrombophilic mutations were analyzed in our study: Factor V Leiden (G1691A) mutation, Prothrombin gene (G20210A) mutation, Methylenetetrahydrofolate Reductase (MTHFR) polymorphisms (C677T and A1298C), Protein C deficiency, Protein S deficiency, Antithrombin III deficiency, Factor XIII mutation, Plasminogen Activator Inhibitor-1 (PAI-1) polymorphism, and ACE I/D polymorphism. Their respective frequencies are now detailed in the manuscript. (lines 136-141)
Comment 4: There is no indication of concomitant therapy (if women were at high risk for thrombophilia, a justification is needed for why they did not receive anticoagulant therapy).
Response: We have now clarified whether participants were receiving anticoagulant therapy and provided justification for those who were not. (lines 120-122)
Comment 5: Information from the high-risk group on preeclampsia and fetal growth restriction would be interesting.
Response: We acknowledge the importance of this information. However, preeclampsia and fetal growth restriction typically develop after week 20, whereas our study focuses exclusively on first-trimester markers. While we have collected data on these conditions, they will be analyzed and presented in a subsequent publication dedicated to the second trimester. Given the limited research on this topic, our team decided to systematically examine and publish each phase of the study separately to provide a clearer understanding of the findings at each stage of pregnancy. However, as preeclampsia and fetal growth restriction typically manifest after week 20, our study focused exclusively on first-trimester markers. While we have collected data on these conditions, they will be analyzed and presented in a subsequent publication dedicated to the second trimester. Given the scarcity of research on this topic, our team considered it essential to systematically study and publish each phase of the research separately to provide a clearer understanding of the findings at each stage of pregnancy.
Comment 6-8: Clarifications regarding figures 1 and 2 (units of measurement, percentages, group labels).
Response: We have revised the figure legends to ensure clarity regarding units of measurement and group names.
Comment 9: When conducting a linear correlation, the authors compare quantitative indicators with qualitative characteristics, which is statistically incorrect.
Response: We acknowledge this issue and have revised our statistical approach to ensure appropriate comparisons in table 2.
Comment 10: The application of a multivariate regression model is mentioned, but results are not adequately presented.
Response: We have now included the multivariate regression results in the appropriate section of the manuscript. (lines 344-375)
We thank both reviewers for their constructive comments and suggestions, which have significantly improved the manuscript. We hope that our responses and revisions adequately address the concerns raised.

Round 2
Reviewer 2 Report
Comments and Suggestions for Authors The authors contributed to the publication all comments made by the reviewer. Personally, I have no new comments. The work may be published with these changes.